# Improved Survival in Hepatocellular Carcinoma Patients with Cardiac Arrhythmia by Amiodarone Treatment through Autophagy

**DOI:** 10.3390/ijms20163978

**Published:** 2019-08-15

**Authors:** Sheng-Teng Huang, Wei-Fan Hsu, Hung-Sen Huang, Jia-Hau Yen, Mei-Chen Lin, Cheng-Yuan Peng, Hung-Rong Yen

**Affiliations:** 1Department of Chinese Medicine, China Medical University Hospital, Taichung 404, Taiwan; 2School of Chinese Medicine, China Medical University, Taichung 404 Taiwan; 3Cancer Research Center for Traditional Chinese Medicine, Department of Medical Research, China Medical University Hospital, Taichung 404, Taiwan; 4Chinese Medicine Research Center, China Medical University, Taichung 404, Taiwan; 5Research Center for Chinese Herbal Medicine, China Medical University, Taichung 404, Taiwan; 6An-Nan Hospital, China Medical University, Tainan 404, Taiwan; 7Division of Hepatogastroenterology, Department of Internal Medicine, China Medical University Hospital, Taichung 404, Taiwan; 8Management Office for Health Data, China Medical University Hospital, Taichung 404, Taiwan; 9Research Center for Traditional Chinese Medicine, Department of Medical Research, China Medical University Hospital, Taichung 404, Taiwan

**Keywords:** amiodarone, autophagy, hepatocellular carcinoma, National Health Insurance Research Database

## Abstract

Hepatocellular carcinoma (HCC) is a primary malignancy of the liver. In basic studies, the regulation of autophagy has offered promising results for HCC treatment. This study aimed to address the question of whether amiodarone can improve survival rates in HCC patients associated with autophagy. Using datasets from the National Health Insurance Research Database, we enrolled patients over 18 years of age that had been diagnosed with HCC between January 1997 and December 2010. Amiodarone and non-amiodarone users were matched at a 1:1 frequency, according to all variables. Additionally, HepG2 cells treated with amiodarone were evaluated by cell viability and autophagic change. Autophagic signaling was examined by immunoblotting and tissue array immunohistochemistry. Of the 10,946 patients diagnosed with HCC, each cohort included 221 patients after 1:1 propensity score matching. The median survival was 36.70 months for the amiodarone users, and 24.48 months for the non-amiodarone users. After adjusting for age, gender, comorbidities and treatment, amiodarone users had a significantly lower risk of mortality. Amiodarone users also demonstrated an improved 3-year survival rate. Furthermore, amiodarone treatment-induced autophagy in HepG2 cells was demonstrated by autophagosome formation associated with increasing LC3B-II, P62, and Beclin-1 expression. Autophagic flux also increased following amiodarone treatment with bafilomycin A1. SiRNA of LC3B knocked down endogenous LC3B formation and restored HepG2 cell viability. This study provides epidemiologic evidence that amiodarone via autophagic degradation machinery may offer survival benefits for HCC patients with a history of arrhythmia. Further randomized, blinded, and placebo-controlled trials are warranted for patients with HCC.

## 1. Introduction

Hepatocellular carcinoma (HCC) is a major public health issue affecting populations globally [1]. In clinical practice, several therapeutic interventions are available to patients. While liver transplantation may provide positive outcomes for selected patients [2], the shortage of liver donors remains a limitation to this solution. Hepatic resection and radiofrequency ablation (RFA) are other possible methods for curative intervention, however tumor recurrence is a major issue after curative hepatectomy for HCC [3,4]. Transarterial embolization (TAE) may be used for intermediate stages of HCC to improve survival [5], and radioembolization with yttrium-90 is another alternative treatment choice [6]. Meanwhile, sorafenib, a multitargeted tyrosine kinase inhibitor, has also been shown to prolong median survival (2.8 months) in the Sorafenib HCC Assessment Randomized Protocol (SHARP) study [7], and has become the standard of care in patients with advanced stage HCC [8]. However, availability of less intrusive oral medications to treat HCC remains limited.

Recent research into the mechanisms involved in cancer development has focused on various implicated pathways, and more specifically on the cellular process of autophagy. Autophagy is a normal physiological process allowing for the lysosomal turnover of cellular energy metabolites, including degradation, renewal of energy, and quality control of intracellular organelles and proteins. Previous studies have shown that several types of cancers, such as breast, ovarian, and prostate cancers, are associated with autophagic dysfunction [9]. In a recent study, amiodarone, an anti-arrhythmic medication, was found to induce autophagy in cellular and animal models of hepatitis B virus (HBV)-related HCC, and suppress liver tumors, suggesting a potential new therapy for HCC [10]. Moreover, a separate study also demonstrated that amiodarone induced autophagy through increased autophagosome formation and degradation, significantly improved liver regeneration, survival rates, and decreased liver injury after 90% massive hepatectomy [11]. However, study results regarding autophagy induction or suppression of HCC growth are conflicting. As an illustration of this conflict, a study on the very early stages of HCC development, by employing a resistant-hepatocyte rat model, found that treatment with amiodarone caused a marked increase in hepatocarcinogenesis [12].

In Taiwan, the compulsory National Health Insurance (NHI) program was launched in 1995, and as of 2015, covered 99.6% of the population [13]. A previous study using the National Health Insurance Research Database (NHIRD) provided by the National Health Insurance Administration and National Health Research Institutes in Taiwan found that patients treated with amiodarone may be associated with an increased risk of cancer development [14]. Arrhythmia is often a comorbidity of patients with HCC, and patients with HCC have a poor prognosis with comorbidity of atrial fibrillation (AF) [15]. Amiodarone is a commonly prescribed class III antiarrhythmic agent for HCC patients with a history of arrhythmia [16]. However, as no large-scale study has examined the interactions between amiodarone and HCC in clinical settings, further research is warranted.

The aim of this study is to evaluate the clinical effects of amiodarone on patients with HCC associated with a history of arrhythmia, and to use a nationwide database in order to bridge the gap between basic studies and clinical trials. We further examine whether amiodarone could be applied as an autophagic inducer to act as a novel adjunct to conventional HCC therapy.

## 2. Results

### 2.1. Demographic Characteristics

Of the 10,946 patients diagnosed with HCC between 1997 and 2010, each cohort group contained 221 patients with a history of arrhythmia after matching for age, sex, comorbidity, treatment, and index year, as shown in Figure 1. The median age was 72.58 (67.98–77.65, 25th and 75th percentile) years for non-amiodarone users, and 73.31 (68.30–77.57) years for amiodarone users. The proportion of males and females for both groups was 61.99% and 38.01%, respectively. The median follow-up periods for the non-amiodarone and amiodarone users were 17.64 (5.55–37.84) months and 27.30 (13.01–45.63) months, respectively. Due to matching comorbidities between the two groups, HBV and HCV were represented in the same proportion and detected in 27.6% and 53.39% of patients in both groups, respectively. Other liver and therapeutic comorbidities, such as liver cirrhosis, ascites, hepatectomy, and RFA or PEI, are shown in Table 1.

### 2.2. The Cause-Specific Hazard Ratio and 3-Year Cumulative Survival of HCC Patients

In Table 2, we demonstrated the univariate and multivariate Cox proportional hazard models for amiodarone users versus non-amiodarone users among HCC patients. Significant crude HRs for mortality in the Cox proportional hazard model were HBV (HR: 0.75, 95% CI: 0.57–0.98, *p* = 0.004), HE (1.65, 1.05–2.58, *p* = 0.0286), ascites (1.86, 1.47–2.36, *p* < 0.0001), RFA or PEI (0.6, 0.45–0.81, *p* = 0.0008), and TAE (0.67, 0.53–0.85, *p* = 0.001). In adjusted model 1, the significant adjusted HRs were amiodarone users (0.77, 0.61–0.97, *p* = 0.0294), RFA or PEI (0.65, 0.48–0.88, *p* = 0.0051) and TAE (0.71, 0.55–0.91, *p* = 0.0079). In adjusted model 2, the significant adjusted HRs were amiodarone users (0.75, 0.59–0.95, *p* = 0.0155), ascites (2.24, 1.73–2.9, *p* < 0.0001), RFA or PEI (0.64, 0.47–0.87, *p* = 0.042), and TAE (0.6, 0.46–0.78, *p* < 0.0001).

The median survival was 36.70 months for the amiodarone users and 24.48 months for the non-amiodarone users; however, the crude HR for the amiodarone users (0.83, 95% CI: 0.66–1.04, *p* = 0.11) did not reach statistical significance (Table 2). After adjusting for age, gender, and co-morbidities, amiodarone users had a significantly reduced risk of mortality (HR: 0.77, 0.61–0.97, *p* = 0.0294 in model 1; HR: 0.75, 0.59–0.95, *p* = 0.0155 in model 2). Amiodarone users also demonstrated an improved 3-year survival with K–M survival analysis (Figure 2A). Patients who cumulatively used amiodarone for more than 84 days showed a greater protective effect than those who used amiodarone for less than 83 days, or not at all (adjust HR: 0.40, 0.27–0.58, *p* < 0.0001) (Table 3); while K–M survival analysis also showed similar results (Figure 2B). From the baseline of the subgroup analysis, more prolonged use of amiodarone demonstrated a notable benefit for those aged over 65 years, with non-HBV, non-HE, non-EGV, and non-therapeutic interventions for HCC (Table 3). 

In order to confirm the protective effect of amiodarone in HCC patients with cardiac arrythmia, we created a separate cohort composed of arrhythmia patients without HCC, and classified into amiodarone users and non-amiodarone users. As shown in Appendix A, arrhythmia patients with amiodarone had significant risk of death (HR: 1.15, 1.12–1.17, *p* = 0.01); however, there was no significant risk of death after adjustment in model 1 (HR: 1.13, 0.97–1.26, *p* = 0.08) or model 2 (HR: 1.05, 0.90–1.34, *p* = 0.10) compared to the non-amiodarone cohort. Additionally, we found that arrhythmia patients with amiodarone treatment had a significantly lower survival rate in comparison with patients without amiodarone treatment (*p* = 0.01) (Appendix A), indicating that amiodarone had no protective effect to prolong the survival rate in arrhythmia patients without HCC. In contrast to HCC patients with cardiac arrhythmia, our results demonstrated that the effect of amiodarone might increase the 3-year survival rate due to another potential mechanism.

### 2.3. Amiodarone Induced HepG2 Cell Death via Autophagic Activation

In order to further validate and clarify the possible reasons behind the prolonged survival rates of HCC patients with amiodarone treatment, we used the HepG2 cell model. Initially, we used the tissue microarray to identify the differences of autophagic markers, including LC3B and P62, between HCC and normal hepatic tissues. We found that the low expression of LC3B was positively correlated with late stage HCC compared with normal liver tissues and early stage HCC, but not associated with P62 expression between normal liver tissues and HCC (Figure 3A,B). However, amiodarone at a concentration of more than 100 µM significantly decreased the HepG2 cell proliferation (Figure 4A). AVO formation (autophagosomes and autolysosomes) is also a distinctive marker for autophagy. Notably, AVO-positive cells, indicating autophagy, were induced after amiodarone treatment (Figure 4B); while LC3B puncta formation in HepG2 cells observed under confocal microscope was also increased (Figure 4C). Additionally, the treatment of amiodarone indeed caused the increase of protein levels, including LC3B II, P62, and Beclin-1 (Figure 5A). To further examine whether LC3B is an essential protein to induce autophagy, we used LC3B siRNA to knock down expression of endogenous LC3B in HepG2 cells. Our results demonstrated that transfection of the LC3B-siRNA significantly inhibited endogenous LC3B protein expression induced by amiodarone compared to cells treated with amiodarone with scrambled siRNA (Figure 5B). The treatment with amiodarone resulted in a significant accumulation in LC3B-II levels in the presence of bafilomycin A1, demonstrating increased autophagic flux (Figure 5C). Moreover, the treatment of amiodarone (100 μM) with LC3B-siRNA effectively recovered HepG2 survival instead of scramble control (Figure 5D). We also found that there was no cytotoxicity presented in normal rat hepatocyte CRL 1439 cells treated with various concentrations of amiodarone (Appendix A). Autophagic markers including LC3B, P62, and Beclin-1 were not activated, although LC3B was significantly down-regulated at concentrations of more than 50 μM amiodarone (Appendix A). Collectively, these findings indicated that amiodarone might activate the autophagic process to inhibit proliferation of liver cancer cells bot not normal hepatocyte.

## 3. Discussion

This is the first large-scale study to investigate the protective effects of amiodarone in patients with HCC associated with a history of arrhythmia. Although the crude HR for the amiodarone users did not reach statistical significance, amiodarone users demonstrated significantly reduced risk of mortality after adjusting for age, gender, and co-morbidities. Notably, amiodarone users, especially those patients using amiodarone for more than 84 days, exhibited improved 3-year survival rates according to K–M survival analysis.

Although several therapeutic modalities have demonstrated improved survival rates for HCC patients [3], sorafenib remains the sole approved oral agent for treatment of the intermediate and advanced stages of HCC [7,17]. While studies have investigated sorafenib combination therapies, such as sorafenib concomitant with transarterial chemoembolization [18] or RFA [19], or immunotherapy which is another potentially promising field of treatment currently under investigation [20], available oral medications for HCC are still limited. Moreover, the adverse effects of sorafenib including adverse skin (hand-foot-skin) reactions, GI upset and asthenia, may limit its application in clinical practice [7,21]. As reported [22], HCC patients with the comorbidity of atrial fibrillation exhibited significantly decreased survival time, as tumor rupture was relatively common among patients with both HCC and AF. This indicates that HCC patients also suffering from arrhythmia would have shortened survival compared with non-arrhythmia patients. Thus, amiodarone may offer an alternative treatment modality to improve survival for hepatoma patients, especially those with a history of arrhythmia.

The cellular process of autophagy is a relatively new concept to clinicians [10,23]. The present investigation into the significance of the autophagic process was inspired by Lan’s study of HBV-related HCC [10]. According to this study, amiodarone might have the liver tumor suppression for hepatoma patients with HBV infection. However, a recent study reported results contradictory to Lan’s study, where amiodarone caused a marked increase in size in cytokeratin-19-positive preneoplastic lesions in a rat model [12]. As such, the different biological effects in distinct models require further elucidation. By contrast, a separate study has reported that chloroquine, a late-stage inhibitor of autophagy, combined with a water-soluble curcumin analog, enhanced apoptosis and cytotoxicity of hepatoma cells, subsequently reducing tumor size [24]. The seemingly conflicting results of these studies may be the result of animal HBV infection, as Lan showed that HBV had a significant correlation with Atg5 and microRNA-224 expression [10]. Moreover, amiodarone repressed miR-224 and increased its target COX-2 expression, to lead to HeLa cell death through SRSF3-PTC induction and miR-224 reduction in a p53-independent manner [25]. Wu et al. reported that a converse correlation between low autophagic activity and high CCND1 expression in HCC recruited from 147 tumor tissue samples from HCC patients and three murine models was noticed. It was considered that amiodarone effectively inhibited tumor growth in orthotopic liver tumor and subcutaneous tumor xenograft models [26]. These findings suggest that autophagy may be another novel avenue for cancer therapy [27]. Thus, we propose that amiodarone could be an autophagic inducer to prolong the 3-year survival rate in patients with hepatoma. Our data also indicated that late stage of HCC indeed had lower expression of autophagic marker of LC3B instead of normal hepatic tissue and early stage of HCC and inhibited the HepG2 cell proliferation by activation of LC3B through autophagic machinery but not for normal hepatocyte. The possible mechanism induced by amiodarone could be due to induced phospholiposis by activation of transcription factor EB (TFEB) which is a critical regulator of the autophagic pathway [28]. Furthermore, the present study fits the niche to address this issue and demonstrates a reduced risk of mortality in HCC patients treated with amiodarone. Of note, short-term use of amiodarone is most commonly prescribed for critical AF patients [29]. However, the adverse effects of long-term amiodarone use, including pulmonary toxicity [30], thyroid dysfunction [31], and optic neuropathy [32], are factors which will require consideration in future trials. Thus, the feasibility of a randomized prospective clinical trial seems questionable given the numbers of patients with long-term medication.

There are several caveats to the present study which must be mentioned. First, this study included a relatively small number of patients, as less than 5% of hepatoma patients in Taiwan have AF or arrhythmia comorbidities, therefore only 221 matching patients with HCC were diagnosed with a history of arrhythmia [15]. Furthermore, database limitations meant that we did not examine other HCC-related liver and tumor characteristics, such as liver aminotransferase, tumor markers [33,34], Child-Pugh classification, size and stage of HCC [3,35], or patients’ performance. In addition, as 44.34% of the included patients received TAE and approximately 23% of the patients received hepatectomy, RFA or PEI, we hypothesized that most of the included patients belonged to the Barcelona Clinic Liver Cancer staging classification A or B [36], and the median survival was longer for patients with HCC receiving TAE [37]. Finally, there are inherent limitations associated with retrospective cohort studies. 

## 4. Materials and Methods

### 4.1. Data Sources

Taiwan implemented the NHI program, a compulsory insurance system in 1995, and currently provides coverage to over 99% of the 23.03 million residents of Taiwan. The NHIRD consists of information on patients’ date of birth, sex, all records of outpatient visits and hospitalizations, details of prescriptions such as prescribed drugs, dosages, and expenditure amounts and diagnosed diseases coded according to the International Classification of Diseases, Ninth Revision, Clinical Modification (ICD-9-CM). 

For the present study, we collected patient information through the Registry for Catastrophic Illness Patients Database (RCIPD) of the NHIRD, which included the entire NHI records for all patients with HCC in Taiwan. The RCPID covers all patients with catastrophic illnesses, including patients with HCC, and these patients are free of co-payment for catastrophic illness-related treatment in Taiwan. The diagnosis of illness is required to be confirmed by pathological, laboratory, and clinical diagnoses by specialists, and undergo regular review by the National Health Insurance Administration. Therefore, the RCIPD has an accurate diagnosis and complete coverage history of patients with HCC. The NHIRD and its corresponding datasets have previously been used for high-quality epidemiologic research [38,39,40,41].

### 4.2. Ethical Approval and Informed Consent

The NHIRD is authorized for use by scientists for research and medical purposes in Taiwan. This study was approved by the Research Ethics Committee of China Medical University and Hospital with certification number CMUH104-REC2-115(CR-2). This study was conducted in accordance with the 1975 Helsinki Declaration. The National Health Insurance Administration scrambled data that could be used to identify individuals or care providers before being sent to the National Health Research Institute for database construction, and is further encrypted before being released to qualified researchers. Researchers who fulfill the requirements are eligible to apply for access to the NHIRD. The database is maintained by and deposited in the National Health Research Institute (http://nhird.nhri.org.tw/en/). It is impossible to identify individuals or care providers by any means in the database. The Research Ethics Committee of China Medical University and Hospital waived informed consent.

### 4.3. Hepatocellular Carcinoma Population

The study population included all patients, aged 18 and over, diagnosed with HCC (ICD-9-CM: 155.0) who also had a medical record of arrhythmia (ICD-9-CM: 427.x) between January 1997 and December 2010, with a follow-up time defined as December 31, 2011. The new diagnosis date of HCC was defined as the index date for the study. Those patients who had previously received amiodarone, or who had withdrawn from the NHI program within a year of follow-up, were excluded from the analysis.

### 4.4. Amiodarone Use

Amiodarone prescriptions were checked for patients with HCC in the outpatient and inpatient database from the index date before the date of mortality, or the end of follow-up. Patients using amiodarone for 28 days or more were defined as amiodarone users; whereas those who had not used amiodarone or had used amiodarone for less than 28 days were defined as non-amiodarone users. Moreover, to observe a dose-response relationship, we further grouped amiodarone users into two groups: one group used amiodarone for 28–83 days (4–12 weeks), and the other used amiodarone for more than 84 days (≥12 weeks).

### 4.5. Study Outcome

The primary outcome was all-cause mortality during the 14-year follow-up and 3-year Kaplan–Meier (K–M) survival. The patients’ dates of death were determined from the registry for catastrophic illness of the NHIRD. All eligible patients were followed-up from the index date to the death of the patient, withdrawal from NHI, or the end of 2011, whichever occurred first.

### 4.6. Covariate Assessment

Information on individual characteristics, age, sex, comorbidities, and HCC-related therapies was recorded. Patient age was classified into two categories (18 to 64 years, ≥65 years). Comorbidities included HBV, viral hepatitis C (HCV), liver cirrhosis, hepatic encephalopathy (HE), ascites, and esophageal or varices (EGV). HCC-related therapies included surgical hepatectomy, RFA, percutaneous ethanol injection (PEI), and TAE. Due to the fact that only two arrhythmic patients from the NHIRD with HCC used sorafenib during the study period, this therapeutic variable was not included in this study. Of note, because HE, ascites, and EGV were complications of liver cirrhosis, we used two adjusted models to avoid confounding. In model 1, all variables were adjusted except for complications associated with liver cirrhosis: HE, ascites, and EGV. In model 2, all variables were adjusted except for liver cirrhosis.

### 4.7. Cell Culture

The HepG2 cells (human hepatocacinoma, BCRC; Bioresource Collection and Research Center, Taiwan) and CRL 1439 cells (normal rat hepatocyte, ATCC, USA) were grown in DMEM complemented with 10% FBS and antibiotics at 37 °C in a 5% CO2 incubator. The cells were treated with different doses of amiodarone for 24 h to examine whether amiodarone could induce autophagy.

### 4.8. Cell Viability Assay

Cells with or without amiodarone treatment for 24 h were washed once with PBS, followed by the addition of 1 mL DMEM containing 0.05 mg/mL 3-(4-,5-dimethylthiazol-2-yl)-2, 5-diphenyltetrazolium bromide (MTT). After incubation at 37 °C for 1 h, the media was removed and the formazan crystals in the cells were solubilized in 1 mL DMSO for optical density (OD) reading at 570 nm using a spectrophotometer.

### 4.9. Quantification of Acidic Vesicular Organelles (AVO) Development

The cells were stained with acridine orange, the cytoplasm and nucleolus fluoresced bright green and dim red, respectively, whereas acidic compartments shone bright red. The prevalence of autophagic cells was also measured by evaluation of the development of AVOs, a marker of autophagy. Cells were stained with acridine orange, removed from the plate with trypsin-EDTA, and collected in phosphate buffer saline (PBS) containing 10% fetal bovine serum. Green and red fluorescence emissions from 10^4^ cells illuminated with blue (488 nm) excitation light were measured with a BD Biosciences FACScan system (San Jose, CA, USA) using CellQuest software. The red/green fluorescence ratio for individual cells was calculated and statistically analyzed.

### 4.10. Immunofluorescence Microscopy

Cells were seeded 5 × 10^5^ cells/well in 12-well plates. For immunostaining, cells were fixed in 4% paraformaldehyde for at least 10 min. Cells were then blocked with 3% BSA in PBS at 37 °C for 1 h and incubated with primary antibody (antibody diluted in 3% BSA) at 4 °C overnight. Cells were then washed 3 times with PBS and incubated with secondary antibody at 37 °C for 1 h in the dark. The cells were washed 3 times with PBS and stained with DAPI, air-dried at 4 °C in the dark, and then washed 3 times with PBS. After mounting in fluoromount media (Sigma-Aldrich Co. LLC, St. Louis, MO, USA), the slides were visualized under a confocal microscopy.

### 4.11. Western Blot Analysis

HepG2 and CRL 1439 cells cultured with or without amiodarone were harvested and total cell protein was extracted using whole cell lysis buffer. The protein concentrations were determined by the Bradford method (Bio-Rad, Berkeley, CA, USA). Samples with equal amounts of protein were subjected to 8–15% sodium dodecyl sulfate polyacrylamide gel electrophoresis (SDS-PAGE) and transferred onto a polyvinylidenedifluoride (PVDF) (Millipore, Bedford, MA, USA) membrane. The membrane was incubated at room temperature in blocking solution (5% nonfat milk) for 1 h followed by incubation for 2 h in blocking solution containing an appropriate dilution of anti-LC3B, -P62, -Beclin-1 antibody (Cell Signaling Technology, Danvers, Massachusetts, USA). After washing, blots were then probed with appropriate secondary horseradish peroxidase (HRP)-conjugated secondary antibodies (Jackson ImmunoResearch, West Grove, PA, USA), detected by an ECL detection system (Millipore, Burlington, Massachusetts, USA), and scanned by MultiGel-21 (Top Bio, Taipei, Taiwan). β-actin was indicated as an internal control. 

### 4.12. Transfection Experiments

Small interfering RNA (SiRNA) targeting the LC3B cDNA sequence and a control SiRNA cDNA sequence from Neon Transfection System (Thermo Fisher Scientific, MPK10096, Waltham, Massachusetts, USA) was applied in this study. HepG2 cells were plated in 6-well plates at densities of 5 × 10^6^/well. The cells were resuspended with the electroporation buffer R, according to the manufacturer’s protocol and then gently pipetted to a single cell suspension. HepG2 cells in the tube contained LC3B and control SiRNA (200 nM), mixed gently. The electroporation condition was 1200 V, 50 ms, 1 pulse. After electroporation, the cells were seeded into 6 wells and incubated at 37 °C, 5% CO_2_ for 24 h. Then, different concentrations of amiodarone were added to the cells for 24 h.

### 4.13. Tissue Array Immunohistochemistry 

The tissue array sections were purchased from the Biomax company (US Biomax, Inc., BC03116a). The slides were de-waxed in the Trilogy system (Trilogy 3 steps in 1, Cell Marque). After being washed 3 times in PBS, the tissue array sections were incubated in 3% H_2_O_2_/methanol for 30 min, and then washed in PBS, and incubated overnight at 4 °C with anti-LC3B and anti-P62 antibodies (cell signaling technology, Danvers, Massachusetts, USA) in blocking solution (antibody diluent with background reducing components, Dako). The tissue array sections were exposed to the secondary antibodies. Then we used DAB kit (Cell Marque, Rocklin, California, USA) to activate HRP and counterstain with Mayer’s hematoxylin. After dehydrating the tissue slides with alcohol, coverslips were mounted. All slides were examined under light microscopy.

### 4.14. Statistical Analysis

Continuous variables were reported as the median (25th and 75th percentiles), and categorical variables were reported as numbers and percentages. Differences in proportions and means were assessed using the chi-squared test and t-test. For mortality risk comparisons between the groups, crude and multivariate analyses of adjusted models with the Cox proportional hazard model were used to estimate the hazard ratio (HR) and 95% confidence interval (95% CI). Survival analysis was performed using the K–M analytic method, and significant differences were determined by the log-rank test. Data were also represented as means ± standard deviation (SD) of three independent experiments. One-way ANOVA was calculated when multiple comparisons were measured. All reported *p*-values were obtained from two-sided tests. Statistical significance was set to *p* < 0.05. All analyses were performed using SAS version 9.4 (SAS Institute Inc., Cary, NC, USA).

## 5. Conclusions

This study provides novel evidence that amiodarone may improve survival rates for patients with HCC associated with a history of arrhythmia. As such, amiodarone, as an autophagic inducer, may be an effective adjunct to conventional HCC therapy to inhibit cancer proliferation; however, further randomized, blinded, and placebo-controlled trials for patients with HCC are warranted.

## Figures and Tables

**Figure 1 ijms-20-03978-f001:**
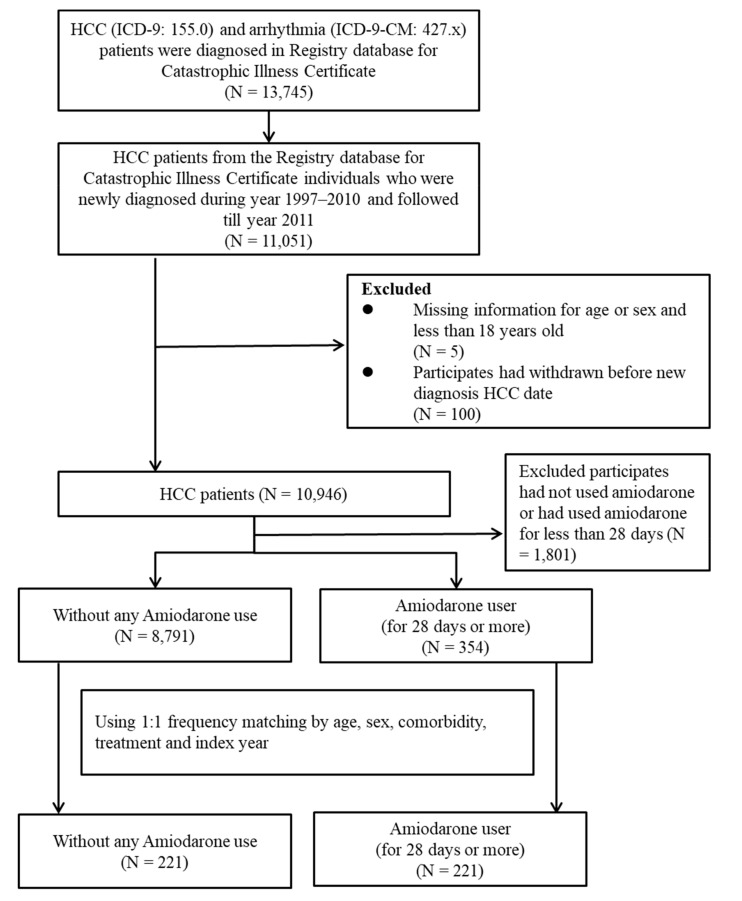
Flow chart of study cases (hepatocellular carcinoma, HCC) from the Longitudinal Health Insurance Database (LHID2000) in Taiwan during 1997–2011.

**Figure 2 ijms-20-03978-f002:**
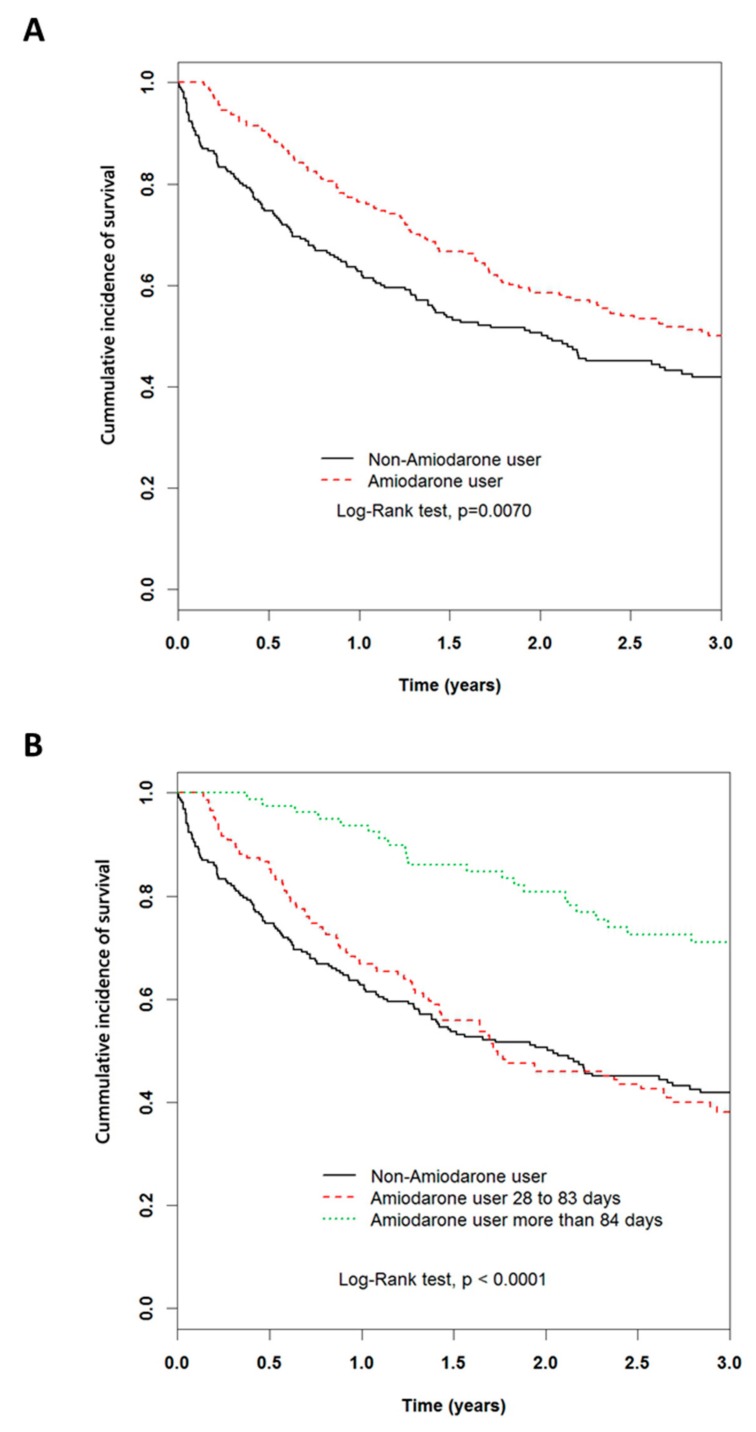
Cumulative incidence of survival in patients with hepatoma with the presence and absence of amiodarone by Kaplan–Meier method analysis. (**A**) 3-year Kaplan–Meier survival curves for amiodarone and non-amiodarone users. (**B**) 3-year Kaplan–Meier survival curves for amiodarone users of more than 84 days, 28–83 days, and non-amiodarone users.

**Figure 3 ijms-20-03978-f003:**
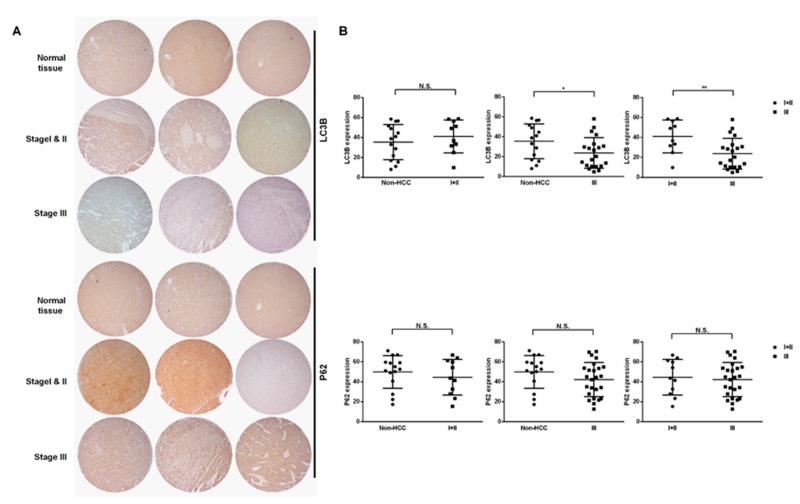
Low LC3B expression correlated positively with late stage hepatocellular carcinoma. (**A**) The expression levels of LC3B and P62 in stages I + II (*n* = 10) and III (*n* = 21) of tumor tissues and adjacent tissues (*n* = 15) were detected by IHC. (**B**) Means of the measurements are shown with black lines. (NS: non-significant difference, * *p* < 0.05, ** *p* < 0.01 compared with untreated normal tissues).

**Figure 4 ijms-20-03978-f004:**
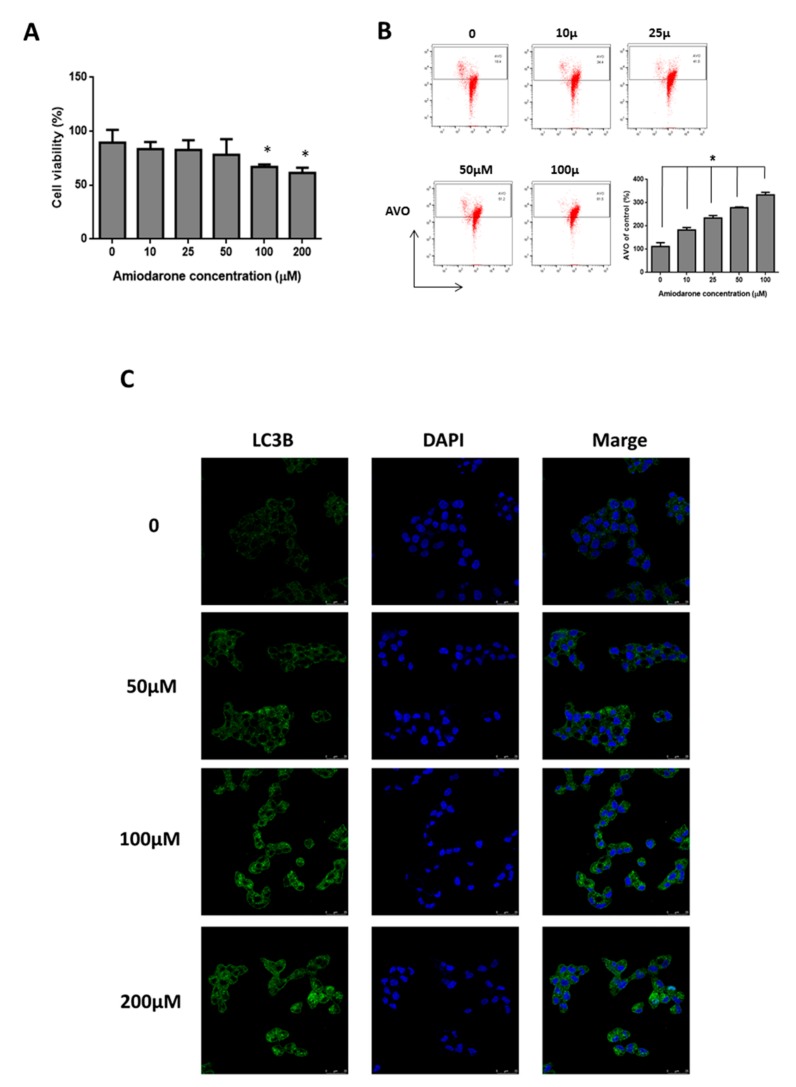
Amiodarone inhibited HepG2 cell growth through autophagy. (**A**) MTT assay for cell viability of HepG2 cells treated with various concentrations of amiodarone for 24 h. (**B**) Quantification of AVO formation in HepG2 cells. AVO induction was measured by acridine orange staining and the percentage of developed AVOs was analyzed by flow cytometry. LC3B puncta formation in HepG2 cells under (**C**) confocal microscope was examined. HepG2 cells were treated with various concentrations of amiodarone, as indicated, and subjected to immunofluorescence analysis of LC3B. Nuclei were stained with DAPI. The results are expressed as the means ± SD from three independent experiments (*n* ≧ 3, * *p* < 0.05 compared with untreated control).

**Figure 5 ijms-20-03978-f005:**
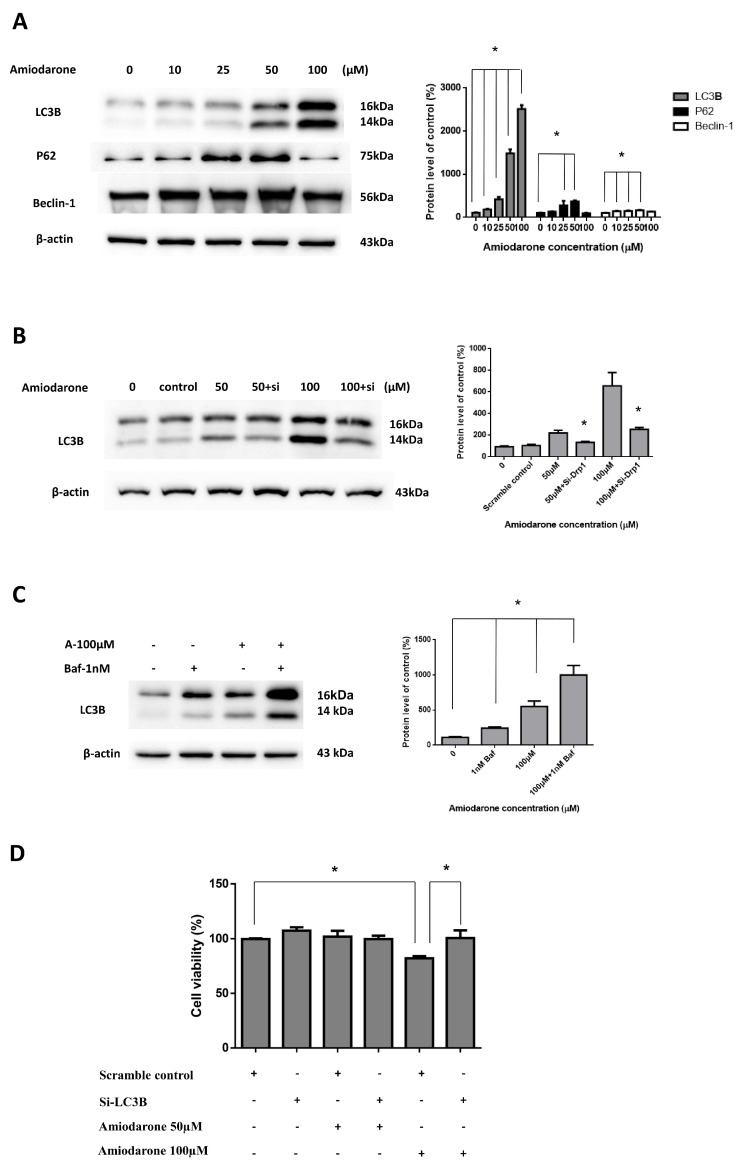
Stimulation of autophagy in HepG2 cells following amiodarone treatment. (**A**) Conversions of LC3B-I to LC3B-II and upregulation of P62 and beclin-1 were detected by immunoblotting following various concentrations of amiodarone treatment in HepG2 cells for 24 h. (**B**) Effect of LC3B-siRNA on the cellular LC3B protein was detected by immunoblotting in HepG2 cells. They were transfected with LC3B and scrambled siRNAs and co-treated with amiodarone (50 and 100 μM) for 24 h. (**C**) HepG2 cells were pretreated with bafilomycin-A1 (1 nM, 2 h) and then co-treated with amiodarone (100 μM) for 24 h to indicate increased autophagic flux. The cells were extracted and the protein levels were measured by immunoblotting. β-actin served as loading control. (**D**) LC3B silencing reversed the reduction of HepG2 cell viability with the treatment of amiodarone. Results are representative of three independent experiments. The results are expressed as the means ± SD from three independent experiments (n ≧ 3, * *p* < 0.05 compared with untreated control).

**Table 1 ijms-20-03978-t001:** Characteristics of arrhythmia patients with HCC according to use of amiodarone.

	Patients with HCC (*N* = 558)				
	Non-Amiodarone Users (*n* = 221)		Amiodarone Users (*n* = 221)		*p*-Value
	n	%	n	%	
**Follow-up time, months (median, 25th and 75th percentile)**	17.64(5.55–37.84)		27.30(13.01–45.63)		
**Gender**					
Women	84	38.01	84	38.01	0.99
Men	137	61.99	137	61.99	
**Age, years**					
18–64	34	15.38	34	15.38	0.99
More than 65	187	84.62	187	84.62	
Median (25th and 75th percentile) ^†^	72.58(67.98–77.65)		73.31(68.30–77.57)		0.7911
**Co-morbidities**					
HBV	61	27.6	61	27.6	0.99
HCV	118	53.39	118	53.39	0.99
Liver cirrhosis	193	87.33	193	87.33	0.99
HE	13	5.88	13	5.88	0.99
Ascites	66	29.86	66	29.86	0.99
EGV	34	15.38	34	15.38	0.99
Hepatectomy	6	2.71	6	2.71	0.99
RFA or PEI	45	20.36	45	20.36	0.99
TAE	98	44.34	98	44.34	0.99

Chi-squared test, ^†^ t-test. Abbreviations; HCC: Hepatocellular carcinoma; HBV: Hepatitis B virus; HCV: Hepatitis C virus; HE: Hepatic encephalopathy; EGV: Esophagogastric varices; RFA: Radiofrequency ablation; PEI: Percutaneous ethanol injection; TAE: Transarterial embolization.

**Table 2 ijms-20-03978-t002:** Cox model with hazard ratios and 95% confidence intervals of mortality associated with amiodarone and covariates among patients with HCC.

Variable	Death no.	Crude ^†^			Adjusted ^‡^-Model 1			Adjusted-Model 2		
	(*n* = 288)	HR	95% C.I.	*p*-Value	HR	95% C.I.	*p*-Value	HR	95% C.I.	*p*-Value
**Amiodarone drug used**										
Non-amiodarone users	143	1.00	reference		1.00	reference		1.00	reference	
Amiodarone users	145	0.83	(0.66–1.04)	0.11	0.77	(0.61–0.97)	0.0294	0.75	(0.59–0.95)	0.0155
**Gender**										
Women	113	1.00	reference		1.00	reference		1.00	reference	
Men	175	0.89	(0.7–1.13)	0.322	0.95	(0.74–1.22)	0.678	0.94	(0.73–1.21)	0.6082
**Age group**										
18–64	38	1.00	reference		1.00	reference		1.00	reference	
More than 65	250	1.31	(0.93–1.85)	0.1175	1.19	(0.82–1.7)	0.3606	1.2	(0.83–1.73)	0.3366
**Co-morbidities**										
**HBV**										
No	218	1.00	reference		1.00	reference		1.00	reference	
Yes	70	0.75	(0.57–0.98)	0.04	0.8	(0.58–1.12)	0.1989	0.79	(0.58–1.08)	0.1406
**HCV**										
No	132	1.00	reference		1.00	reference		1.00	reference	
Yes	156	0.99	(0.79–1.25)	0.9556	0.97	(0.71–1.31)	0.8243	0.9	(0.69–1.17)	0.4343
**Liver cirrhosis**										
No	40	1.00	reference		1.00	reference				
Yes	248	0.74	(0.53–1.04)	0.0821	1.09	(0.7–1.69)	0.7064			
**HE**										
No	267	1.00	reference					1.00	reference	
Yes	21	1.65	(1.05–2.58)	0.0286				1.42	(0.89–2.28)	0.1448
**Ascites**										
No	173	1.00	reference					1.00	reference	
Yes	115	1.86	(1.47–2.36)	<0.0001				2.24	(1.73–2.9)	<0.0001
**EGV**										
No	237	1.00	reference							
Yes	51	1.15	(0.85–1.56)	0.3549				0.94	(0.68–1.31)	0.7245
**Hepatectomy**										
No	281	1.00	reference		1.00	reference		1.00	reference	
Yes	7	0.48	(0.21–1.08)	0.0746	0.57	(0.25–1.31)	0.1866	0.66	(0.29–1.51)	0.3191
**RFA or PEI**										
No	231	1.00	reference		1.00	reference		1.00	reference	
Yes	57	0.6	(0.45–0.81)	0.0008	0.65	(0.48–0.88)	0.0051	0.64	(0.47–0.87)	0.0042
**TAE**										
No	155	1.00	reference		1.00	reference		1.00	reference	
Yes	133	0.67	(0.53–0.85)	0.001	0.71	(0.55–0.91)	0.0079	0.6	(0.46–0.78)	<0.0001

Crude HR ^†^: Represented relative hazard ratio; Model 1: Adjusted HR^‡^ represented adjusted hazard ratio, mutually adjusted for Amiodarone drug used, age, gender, HBV, HCV, and liver cirrhosis in Cox proportional hazard regression; Model 2: Adjusted HR† represented adjusted hazard ratio, mutually adjusted for Amiodarone drug used, age, gender, HBV, HCV, HE, ascites, EGV, and treatment in Cox proportional hazard regression; Abbreviations: HCC: Hepatocellular carcinoma; HBV: Hepatitis B virus; HCV: Hepatitis C virus; HE: Hepatic encephalopathy; EGV: Esophagogastric varices; RFA: Radiofrequency ablation; PEI: Percutaneous ethanol injection; TAE: Transarterial embolization.

**Table 3 ijms-20-03978-t003:** Hazard ratios for duration of amiodarone use and mortality in arrhythmic patients with HCC.

Variables	Non-User (*n* = 221)	Users (more than 28 days)		*p* for Trend ^†^
		28 to 83 days (*n* = 142)	More than 84 days (*n* = 79)	
	aHR	aHR (95% CI)	aHR (95% CI)	
**Total**	1.00	1.02(0.79–1.32)	0.40(0.27–0.58) ***	<0.0001
**Gender**				
Women	1.00	1.03(0.67–1.6)	0.49(0.29–0.83) **	0.0156
Men	1.00	1.00(0.73–1.38)	0.3(0.17–0.52) ***	0.0002
**Age group**				
18–64	1.00	1.66(0.73–3.75)	0.6(0.24–1.53)	0.5497
More than 65	1.00	0.95(0.72–1.25)	0.34(0.23–0.52) ***	<0.0001
**Co-morbidities**				
**HBV**				
No	1.00	0.93(0.7–1.26)	0.32(0.21–0.5 )***	<0.0001
Yes	1.00	1.29(0.76–2.19)	0.73(0.35–1.54)	0.7191
**HCV**				
No	1.00	0.91(0.62–1.34)	0.32(0.19–0.55) ***	<0.0001
Yes	1.00	1.15(0.81–1.62)	0.46(0.27–0.78) **	0.0264
**Liver cirrhosis**				
No	1.00	0.69(0.34–1.42)	0.17(0.06–0.45) ***	0.0003
Yes	1.00	1.06(0.81–1.4)	0.52(0.35–0.78) **	0.0080
**HE**				
No	1.00	1.02(0.78–1.33)	0.37(0.25–0.55) ***	<0.0001
Yes	1.00	1.06(0.34–3.32)	0.82(0.17–3.9)	0.8654
**Ascites**				
No	1.00	1.1(0.79–1.53)	0.37(0.23–0.59) ***	0.0002
Yes	1.00	0.82(0.54–1.24)	0.42(0.22–0.79) **	0.0103
**EGV**				
No	1.00	1.02(0.77–1.35)	0.37(0.24–0.55) ***	<0.0001
Yes	1.00	1.03(0.54–2)	0.63(0.23–1.72)	0.4832
**Hepatectomy**				
No	1.00	1(0.77–1.3)	0.39(0.27–0.57) ***	<0.0001
Yes	1.00	1.35(0.2–9.2)	-	0.7595
**RFA or PEI**				
No	1.00	0.91(0.69–1.21)	0.33(0.21–0.51) ***	<0.0001
Yes	1.00	1.56(0.86–2.85)	0.9(0.39–2.05)	0.7099
**TAE**				
No	1.00	0.9(0.64–1.26)	0.24(0.13–0.44) ***	<0.0001
Yes	1.00	1.12(0.76–1.66)	0.74(0.46–1.21)	0.3662

aHR: Represented adjusted hazard ratio, mutually adjusted for amiodarone, age, gender, HBV, HCV, liver cirrhosis, HE, ascites, EGV, and treatment in Cox proportional hazard regression. **: *p* < 0.01; ***: *p* < 0.001; **^†^** Two-tailed *p*-values of test for linear trend. Abbreviations: aHR: Adjusted hazard ratio; CI: Confidence interval; HCC: Hepatocellular carcinoma; HBV: Hepatitis B virus; HCV: Hepatitis C virus; HE: Hepatic encephalopathy; EGV: Esophagogastric varices; RFA: Radiofrequency ablation; PEI: Percutaneous ethanol injection; TAE: Transarterial embolization.

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
