# Peer review of "Improved Survival in Hepatocellular Carcinoma Patients with Cardiac Arrhythmia by Amiodarone Treatment through Autophagy"

_ijms, 2019, doi:10.3390/ijms20163978_

Round 1

Reviewer 1 Report

In this study, Huang et al. seek the therapeutic effects of amiodarone against HCC using the Taiwanese cohort.

I reviewed the previous version (ijms-521741) and this revised manuscript fixed concerns/questions raised in the previous one. The authors addressed all my comments and I have no further comments.

Reviewer 2 Report

The authors have effectively described in a retrospective analysis, the beneficial effects of using amiodarone in a population of patients with HCC with the comorbidity of cardiac arrhythmia. In these patients, there appears to be a survival advantage.

With that being said, any potential benefit must be balanced against long term side effects of amiodarone therapy, such as chronic pulmonary fibrosis and cardiac toxicities.

Additionally, novel agents such as PDL-1 inhibitors have proved very promising for the treatment of HCC.   Therefore, the probability of a prospective randomized placebo controlled trial trial  occuring are low.

With those caveats, given the potential benefit to a small number of patients with HCC and arrhythmia, this paper may provide an avenue   of treatment worth considering.